# The Gene Expression Profile Differs in Growth Phases of the *Bifidobacterium Longum* Culture

**DOI:** 10.3390/microorganisms10081683

**Published:** 2022-08-21

**Authors:** Vladimir A. Veselovsky, Marina S. Dyachkova, Dmitry A. Bespiatykh, Roman A. Yunes, Egor A. Shitikov, Polina S. Polyaeva, Valeriy N. Danilenko, Evgenii I. Olekhnovich, Ksenia M. Klimina

**Affiliations:** 1Department of Biomedicine and Genomics, Federal Research and Clinical Center of Physical-Chemical Medicine of Federal Medical Biological Agency, Moscow 119435, Russia; 2Department of Biotechnology, Vavilov Institute of General Genetics Russian Academy of Sciences, Moscow 119991, Russia; 3Department of Biological and Medical Physics, Moscow Institute of Physics and Technology, National Research University, Dolgoprudny 141701, Russia

**Keywords:** *Bifidobacterium longum*, RNA sequencing, transcriptome, DEGs, growth phase

## Abstract

To date, transcriptomics have been widely and successfully employed to study gene expression in different cell growth phases of bacteria. Since bifidobacteria represent a major component of the gut microbiota of a healthy human that is associated with numerous health benefits for the host, it is important to study them using transcriptomics. In this study, we applied the RNA-Seq technique to study global gene expression of *B. longum* at different growth phases in order to better understand the response of bifidobacterial cells to the specific conditions of the human gut. We have shown that in the lag phase, ABC transporters, whose function may be linked to active substrate utilization, are increasingly expressed due to preparation for cell division. In the exponential phase, the functions of activated genes include synthesis of amino acids (alanine and arginine), energy metabolism (glycolysis/gluconeogenesis and nitrogen metabolism), and translation, all of which promote active cell division, leading to exponential growth of the culture. In the stationary phase, we observed a decrease in the expression of genes involved in the control of the rate of cell division and an increase in the expression of genes involved in defense-related metabolic pathways. We surmise that the latter ensures cell survival in the nutrient-deprived conditions of the stationary growth phase.

## 1. Introduction

Different categories of bacteria have different requirements for growth conditions. Factors such as oxygen, pH, temperature, and light exert a significant impact on microbial growth. A bacterial population’s generation time varies among species and depends on how well the growth conditions match the bacteria’s requirements. Bacteria often transition between rapid growth (exponential phase) and slow growth (stationary phase), depending on nutrient availability [1,2]. These transitions require that the cells undergo substantial transcriptomic rearrangements. The vast majority of the transcripts that are expressed in the exponential phase are repressed in the stationary phase and many others are activated in the stationary phase. The transition between the exponential and the stationary phases is usually marked by physiological, morphological, and transcriptional differences. Moreover, the stationary phase can be considered as stressful and many of the transcripts that are highly expressed during this phase are stress-related [3]. To date, transcriptomic approaches have been successfully used to study gene expression in different growth phases of bacteria [4,5,6].

The human gut microbiota harbors a complex and dynamic microbial community, which plays a crucial functional role in supporting human health. Bifidobacteria are among the first colonizers of the gut and remain a major component of a person’s healthy gut microbiota throughout his life [7]. Bifidobacteria are gram-positive, high in G+C content, rod shaped, non-motile and non-spore-forming bacteria, belonging to the Actinobacteria phylum; currently the genus Bifidobacterium includes 94 species. Among the bifidobacteria colonizing the human gut, *Bifidobacterium longum* deserves special mention due to its prevalence (and in some cases abundance) in both infants and adults. Despite the dramatic shift in microbiota composition that occurs following the transition from an exclusively milk-based diet to solid foods, it has been demonstrated that specific *B. longum* strains may persist in the intestinal microbiota over time. Otherwise, new strains can also be acquired by the adult host from his diet [8]. Since bifidobacteria are obligate commensals that typically inhabit the intestines of animals, including humans, unlike free-living bacteria, they are adapted to the specific environmental conditions of the intestines. Conditions inside the host’s organism are marked by constant change, which imposes certain challenges on the bacteria living in that ecosystem. The mechanisms that allow commensal bacteria to survive under starvation conditions and resist various stresses while keeping the microbial population in check can be very different from those employed by non-host-inhabiting bacteria and thus are of considerably great interest. In this study, we chose a gut-derived strain of human origin that belongs to the species *B. longum*. The rationale behind this choice was that by studying a bacterium adapted to gut conditions, the transcriptome analysis will reveal how such organisms react to different environmental factors that it encounters in the human organism. Nutrient deprivation as well as physical and chemical stresses result in unbalanced growth and entail many changes at the molecular level [9]. Thus, studying those changes will help us to better understand the mechanisms that promote long-term stability of the composition of the human commensal microbiota.

The state of the microbial population of various types of intestinal microbiota is very dynamic and its quantitative composition changes depending on external conditions, for example, as a result of pathological processes, changes in diet, or as a result of taking antibiotics or other drugs. Therefore, under natural conditions, microbial populations are in a different state, from active replication and division to reproduction deprivation, which is very similar to the growth phases observed in laboratory conditions. Metagenomic approaches have helped to observe microbial species in different growth phases and to estimate bacterial growth dynamics in natural conditions within a microbial community [10]. Of course, laboratory conditions do not ideally simulate the development of a microbial population, and they differ from the native, however, they provide a model for studying the response of bacteria to external factors, such as changes in microbial population density or nutrient availability. Studying how the expression profile of the *B. longum* GT15 strain differs between growth phases will shed light on the metabolic pathways that are affected by alternating favorable and stress conditions. In addition, information about the changes occurring at different phases of the cultivation of microorganisms is important for researchers who study microorganisms mainly in laboratory rather than natural conditions. Therefore, understanding the gene expression patterns of bifidobacteria in different growth phases can also be useful for experimental planning in biotechnology.

An efficient method for revealing growth phase-related metabolic pathways is to use the omics methods. In *Bifidobacterium bifidum* PRL2010, transcriptomic study showed a switch from cell division-related gene expression at the lag phase to a carbohydrate metabolism-related gene repertoire at the exponential growth phase [11]. Combined transcriptome and proteomic studies showed that from the exponential to the stationary growth phase, the switch from glucose fermentation to galactose usage and the switch from homolactic to mixed acid fermentation were both seen in *Lactobacillus rhamnosus* GG [12]. Additionally, Zhang transcriptomic analysis revealed that genes related to nucleotide transport and metabolism, energy production and conversion, and chaperones were primarily expressed during the exponential phase, whereas genes related to carbohydrate metabolism, inorganic ion transport, and chaperones were highly expressed at the stationary phase in *L. casei* [13]. Finally, very recently Wang et al. demonstrated that at the stationary growth phase of *B. animalis* subsp. *lactis* A6, sugars transport and metabolism increased, resulting in an enlarged carbon source using profile to counteract glucose consumption, as well as genes related to the cysteine-cystathionine-cycle (CCC) pathway, glutamate dehydrogenase, branched-chain amino acids (BCAAs) biosynthesis, and Clp protease were all up-regulated in the stationary phase, which may increase the capacity of B. *lactis* A6 to withstand acid during this stage of its life cycle [14]. In this study, we applied the RNA-Seq technique to study the global gene expression in different growth phases of *B. longum* GT15.

## 2. Materials and Methods

### 2.1. Strain and Cell Growth

In this work, we used the *B. longum* GT15 strain whose genome was sequenced and submitted to GenBank and assigned the accession no. CP006741 earlier [15]. The strain *Bifidobacterium longum* subsp. *longum* GT15 was isolated from the feces of a healthy adult inhabiting Central Russia and demonstrated good probiotic properties [9,15] The cultivation of *B. longum* GT15 was carried out in Lactobacillus MRS Broth culture medium (HiMedia, India) supplemented with 0.5% cysteine (HiMedia, India). *B. longum* GT15 was cultured at 37 °C under anaerobic conditions (HiAnaerobic System-Mark III, AnaeroHiGas Pack 3.5 L; HiMedia, India). The cells were sampled at three different time points based on the growth profile of the strain as follows: lag phase (GT15_lag), exponential phase (GT15_ex), and stationary phase (GT15_st). The GT15_lag, GT15_ex, and GT15_st time points were OD~0.1, OD~0.5, and OD~0.9 for *B. longum* GT15. The experiment was carried out in three biological replicates. The cells were treated with RNA Protect Bacteria Reagent (QIAGEN).

### 2.2. RNA Extraction, Library Preparation, and High-Throughput Sequencing

RNA Extraction, library preparation, and high-throughput sequencing were performed as previously described by Veselovsky et al. [16]. Total RNA extraction and purification was performed using the RNeasy Mini Kit (QIAGEN). Removal of residual gDNA was performed using the TURBO DNA-free Kit (Invitrogen, Waltham, MA, USA) and the RNase-Free DNase Set (QIAGEN, Hilden, Germany). The concentration and quality of the extracted RNA were checked using the Quant-iT RiboGreen RNA Assay Kit (Thermo Fisher Scientific, Walthamm, MA, USA) and the Agilent RNA 6000 Pico Kit (Agilent Technologies, Santa Clara, CA, USA), respectively.

Total RNA (2 µg) was used for library preparation. Ribosomal RNA was removed from the total RNA using the RiboZero rRNA Removal Kit (Bacteria) (Epicentre/Illumina, Madison, WI, USA), and libraries were prepared using the NEBNext^®^ Ultra II Directional RNA Library Prep Kit (NEB, Ipswich, MA, USA), according to the manufacturer protocol. Subsequently, RNA cleanup was performed with the RNA°Clean XP kit (Beckman Coulter, Brea, CA, USA). The library underwent a final cleanup using the Agencourt AMPure XP system (Beckman Coulter, Brea, CA, USA) after which the libraries’ size distribution and quality were assessed using a high sensitivity DNA chip (Agilent Technologies). Libraries were subsequently quantified by Quant-iT DNA Assay Kit, High Sensitivity (Thermo Fisher Scientific). Finally, equimolar quantities of all libraries (12 pM) were sequenced by a high throughput run on the Illumina HiSeq using 2 × 100 bp paired-end reads and a 5% Phix spike-in control. Before loading the cBot system, the libraries were incubated at 98 °C for 2 min and then cooled on ice to improve the hybridization of GC-rich sequences. The dataset of RNA-Seq analysis was deposited to the NCBI under the project name PRJNA628664.

### 2.3. Data Processing and Analysis

Quality of the raw reads was assessed via FASTQC v0.11.5. The sequenced reads were mapped to the reference *B. longum* GT15 assembly (GenBank accession no. GCA_000772485.1), using HISAT2 (v2.2.1) [17]. SAMtools (v1.11) software [18] was used to sort and index BAM files [19]. Mapping quality and coverage along genes were assessed with QualiMap (v2.2.2), and individual reports were merged with MultiQC (v1.9) [20,21]. Mapped reads were assigned to genes with featureCounts (v2.0.1) [22]. Differential gene expression analysis was performed using the edgeR (v3.30.3;) package for R [23]. Genes with a false discovery rate (FDR) cutoff of 0.05 and with a logarithmic fold change (log_2_FC) threshold of |1| (i.e., ≥|2|-fold change) were considered to be differentially expressed. Plots were generated within R using ggplot2 (v3.3.2) [24]. DEGs were then subjected to enrichment analysis of COG functions and the Kyoto Encyclopedia of Genes and Genomes (KEGG) pathways. Bedtools genomecov was used for coverage calculation [25]. Using the moving average, coverage distribution tables were obtained. Data visualization was performed using GNU/R [19].

## 3. Results

### 3.1. Differences in Gene Transcription Profiles

To scrutinize the transcriptome patterns of the three growth phases of *B.longum* GT15, we applied the RNA-seq technique for sequencing the whole transcriptome from three independent biological replicates of each growth phase.

A total of 11,994,523, 12,045,570, and 12,513,078 reads were obtained for the samples at the lag phase (GT15_lag), exponential phase (GT15_ex), and stationary phase (GT15_st), respectively. After filtering out poor quality reads, the number of effective reads mapped to the genome of *B. longum subsp. longum* GT15 was reduced to 10,705,476, 10,794,398, and 11,095,975, respectively (Table 1).

Genes that were significantly differentially expressed (log_2_FC ≥ |1|, FDR < 0.05) in each of the growth phases were counted in each of the growth phases: 239 DEGs, in total, were detected in the exponential phase compared to the lag phase, including 141 downregulated and 98 upregulated genes; 553 DEGs were detected in the stationary phase compared to the lag phase among which 277 genes were downregulated and 276 genes upregulated; and 392 DEGs were detected in the stationary phase compared to the exponential phase, including 190 downregulated and 202 upregulated genes (Figure 1, Appendix A).

The expression of 45 DEGs throughout the experiment. A comparison of the expression of these 45 genes (upregulated and downregulated) is presented in Figure 2.

The downregulated genes were classified according to their function as follows: ABC transporters, quorum sensing, metabolic pathways, homologous recombination, histidine metabolism and glycerolipid metabolism. The transporters (BLGT_07630, BLGT_07625, BLGT_07620, BLGT_07635, BLGT_07350, BLGT_06730, BLGT_06725, BLGT_06720, BLGT_05680, BLGT_04630, BLGT_04635, BLGT_07550, BLGT_01180), Enzymes (BLGT_03450, BLGT_09605, BLGT_07635), DNA repair and recombination proteins (BLGT_07210), Glycosyltransferases (BLGT_03450).

The upregulated genes were classified according to their function as follows: metabolic pathways, ABC transporters, RNA degradation, biosynthesis of amino acids, carbon metabolism, pantothenate and CoA biosynthesis, methane metabolism, starch and sucrose metabolism, two-component system, biosynthesis of cofactors, biosynthesis of secondary metabolites, microbial metabolism in diverse environments, glycolysis/gluconeogenesis, and galactose metabolism. Enzymes (BLGT_03060, BLGT_09090, BLGT_03845), Messenger RNA biogenesis and Exosome (BLGT_03845, BLGT_07730), Transporters (BLGT_09100, BLGT_09095).

### 3.2. KEGG Pathway and Functional Annotation of DEGs

The biologically significant DEGs were further analyzed using the Kyoto Encyclopedia of Genes and Genomes (KEGG) pathways. As predicted by KEGG, 50 pathways were altered in GT15_ex as compared to GT15_lag (Figure 3, Appendix A), 75 pathways were altered in GT15_st as compared to GT15_lag (Appendix A, Appendix A), and 75 pathways were altered by GT15_st as compared to GT15_ex (Appendix A, Appendix A). A detailed representation of these pathways is given in Figure 3 and Appendix A, Appendix A and Appendix A.

Analysis of the pathways affected by DEGs revealed a higher number of transcripts of ABC transporters in the lag phase than either of the exponential and stationary phases (Figure 3 and Appendix A). Overall, the number of DEGs in the stationary phase was smaller than that of DEGs in the exponential phase. Similar results were obtained for DEGs linked to glycine, serine, threonine and glycerolipid metabolism. The number of DEGs linked to fatty acid metabolism and fatty acid biosynthesis remain unchanged in both the lag and the exponential phases and decreased in the stationary phase. As for DEGs linked to glycolysis/gluconeogenesis, nitrogen metabolism, and arginine biosynthesis, their number was highest in the exponential phase and lowest in the lag phase. In addition, a number of upregulated DEGs related to methane metabolism were observed in the exponential phase. Only the number of DEGs related to valine, leucine, isoleucine, and histidine metabolism was higher in the lag and stationary phases as compared to the exponential phase.

Functional analysis revealed that DEGs between the growth phases GT15_ex and GT15_lag were divided functionally into 16 groups: Translation, ribosomal structure and biogenesis; transcription; replication, recombination and repair; cell cycle control, cell division, chromosome partitioning; defense mechanisms signal transduction mechanisms; cell wall/membrane/envelope biogenesis; post-translational modification, protein turnover, chaperones; energy production and conversion; carbohydrate transport and metabolism; amino acid transport and metabolism; nucleotide transport and metabolism; coenzyme transport and metabolism; lipid transport and metabolism inorganic ion transport and metabolism; general function prediction only. As for DEGs between the growth phases GT15_st and GT15_lag, they were additionally linked to secondary metabolites biosynthesis, transport and catabolism, cell motility and intracellular trafficking, secretion, and vesicular transport (Figure 4).

Overall, functional analysis demonstrated that the number of upregulated DEGs linked to translation, ribosomal structure, and biogenesis group was higher in GT15_ex than in GT15_lag. At the same time, the number of downregulated DEGs in GT15_st exceeded that of the phase GT15_ex. The most numerous DEGs were linked to defense mechanisms. Although the total number of upregulated and downregulated DEGs for each comparison was approximately the same, the overall trend of the number of DEGs was upward as we moved from the GT15_lag phase to GT15_st. In the GT15_st phase, the number of upregulated DEGs involved in transcription and the number of downregulated DEGs involved in cell wall/membrane/envelope biogenesis increased significantly. The number of upregulated and downregulated DEGs linked to carbohydrate transport and metabolism was increased in the GT15_ex phase as compared to the GT15_lag phase and in the GT15_st phase as compared to the GT15_ex, respectively. The number of upregulated DEGs linked to posttranslational modification, protein turnover, and chaperones also increased in the In GT15_st phase.

## 4. Discussion

The RNA-seq technique has been suggested as one of the best tools for understanding complex biological processes, such as how bifidobacteria efficiently adapt to the human gut. Unfortunately, only a handful of studies used RNA-seq to study bifidobacteria. One study authored by Bottacini et al. recently mapped the genes and operons that are actively transcribed in *B. breve* UCC2003 in the logarithmic phase [26]. Another study explored the functional landscape of gene expression of the gut-derived strain *B. breve* DSM 20213 upon exposure to linoleic acid in vitro [27]. Yang J et al. combined transcriptomics with metabolomics to investigate the utilization and metabolism of xylooligosaccharides in *B. adolescentis* 15703. Transcriptome analysis showed that xylooligosaccharides upregulated the expression of genes linked to ABC transporters, galactosidase, xylosidase, glucosidase, and amylase metabolism [28]. Wei Y et al. studied the adaptive physiological and transcriptional responses of *B. longum* to acid stress, revealing an increase in the expression of genes linked to fatty acid metabolism in the cell membrane [29]. Activation of glutamate metabolism genes in the stationary phase may indicate increased resistance to acids [14]. Transcriptome analysis has also been used to study the effects of oxidative stress on *B. longum subsp. longum* BBMN68, showing a decrease in the expression of genes involved in nucleotide metabolism, amino acid transport, protein, and chaperone metabolism and an increase in carbohydrate metabolism, translation and biogenesis [30]. In the stationary phase, when bacterial growth is suppressed, there is a decrease in the expression of genes responsible for the biosynthesis of peptidoglycans, which are the main component of the cell wall [31].

In this study, we were interested in studying global gene expression at all the growth phases in the strain *B. longum* GT15. The lag phase represents the earliest and most poorly understood phase of the bacterial growth cycle. In the lag phase, ABC transporters are expressed much more actively, which may be explained by an increase in the rate of absorption of nutrients from the medium in preparation for active cell division. In the exponential phase, the expression of genes encoding ABC transporters gradually drops to reach its minimum in the stationary growth phase. The expression of genes encoding ABC transporters is perhaps necessary to maintain the growth rate of the culture. According to COG classification and KEGG protein predicted genes, the exponential phase, compared to the lag phase, is marked by the increased expression of genes involved in the synthesis of amino acids (in particular, alanine, and arginine) and ATP, as well as genes involved in translation, formation of ribosome structures, and biogenesis. Activation of those genes is known to trigger active cell division, leading to exponential growth of the culture. To maintain high rates of cell growth in the exponential growth phase, genes associated with energy metabolism (glycolysis/gluconeogenesis and nitrogen metabolism) are mainly activated. For example, in the stationary phase characterized by slow cell division, the expression of the BLGT_RS09365 gene that encodes a histidine ammonia-lyase (nitrogen metabolism) drops significantly (Figure 2). Histidine ammonia-lyase expression is more than 15 times higher in the growth lag phase relative the stationary growth phase and 7 times higher relative to the exponential growth phase.

In the stationary phase, the expression of genes linked to translation, ribosomal structure, biogenesis, replication, recombination and reparation drops significantly. One example is that of the reduced expression of the BLGT_RS04410 gene (DEAD/DEAH box helicase) (expression decreases by 12.9 in GT15_st compared to GT15_lag), which plays an important role in RNA metabolism [32]. A decline in the expression of genes involved in the metabolic pathways involving cell wall/membrane/envelope biogenesis, fatty acid metabolism and biosynthesis, transport and metabolism of amino acids, such as alanine and tryptophan, as well as the synthesis of Vitamin B6, was also noted. All of these changes in gene expression in the stationary growth phase contribute to the reduction of the cell division rate. Moreover, in the stationary phase, the number of DEGs (both upregulated and downregulated) involved in the pathways linked to defense mechanisms, is much higher compared to the exponential and lag phases. One possible explanation is that the cells experience stress due to a nutrient deficiency, and respond via mobilization of defense mechanisms. In addition, the expression of genes coding for the chaperonin GroEL and the co-chaperone GroES, which account for normal formation of proteins under stress conditions, increase in the stationary phase. These rearrangements of expression indicate the stoppage of active cell division related to nutrient deficiency. This hypothesis is further supported by the fact that the expression of genes involved in carbohydrate (starch and sucrose) transport and metabolism is increased. It is plausible that bifidobacteria restructure their metabolism to be able to search for new energy sources in conditions of nutritional deficiency.

Interestingly, the expression of the BLGT_RS07530 gene (expression is increased by 31.3 in GT15_st compared to GT15_lag), which encodes a protein belonging to the WXG100 family type VII secretion target, was increased in the stationary phase. WXG100 is a superfamily of proteins that are found in *Mycobacterium tuberculosis* and other bacteria. Although the exact function of these proteins has not been established yet, it is possible that they are necessary for the activation of cell-mediated immune responses [33]. The expression of this protein in the case of *B. lon**gum* GT15 can thus activate the host immune response and exert a probiotic effect under stress conditions in general, which are not necessarily limited to nutrient deficiency.

An analysis of *B. lon**gum* GT15 gene expression showed that in the stationary phase, the expression of genes that promote long-term survival under stress is activated. *E. coli* show similar changes in the expression of genes aimed at long-term survival under stress conditions, such as osmotic stress, periplasmic shock, cold shock, and others [4].

It should be noted that the growth phase that is considered natural to bacterial populations is rather the stationary growth phase due to the scarcity of stimulating factors (e.g., nutrients) and the abundance of growth inhibiting factors (e.g., accumulation of toxic by-products, presence of stress factors) [34]. However, the stationary phase is not necessarily the final growth phase. Bacterial growth resumes as soon as environmental conditions become favorable again [6]. Thus, the difference we found in gene expression in the stationary phase and the lag phase provide us with an idea of the regulatory mechanisms of bifidobacteria that stimulate the effective re-growth of culture in vivo.

## 5. Conclusions

Commensal microorganisms in the host organism are constantly in a state of transition from the stationary to the growth phase. We have demonstrated that in the cell culture of *B. longum* GT15, the preparation for cell division causes the expression of genes in the lag phase to increase, some of whose function may be related to active substrate consumption. The synthesis of amino acids, energy metabolism, and translation are all functions of genes activated during the exponential phase, and these processes all encourage vigorous cell division, which results in exponential growth of the culture. In the stationary phase, we found an increase in the expression of metabolic pathways associated to defense and a reduction in the expression of genes involved in the regulation of cell division rate. This latter, we hypothesize, enables cell survival throughout the stationary growth phase in nutrient-poor circumstances.

## Figures and Tables

**Figure 1 microorganisms-10-01683-f001:**
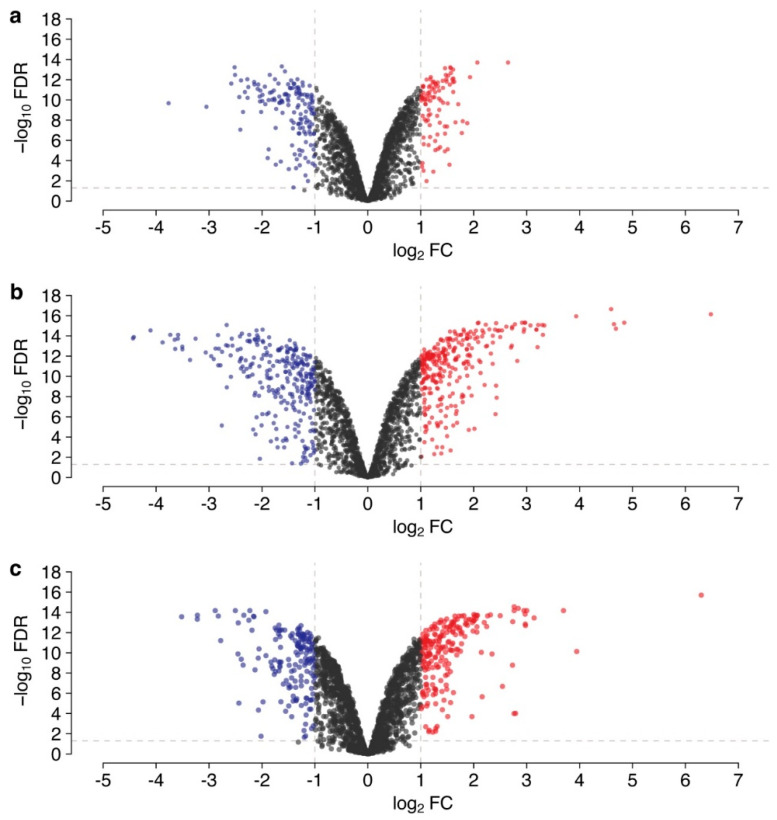
Volcano plot depicting DEGs in *B. longum subsp. longum* GT15 between: (**a**) GT15_ex to GT15_lag, (**b**) GT15_st to GT15_ex, (**c**) GT15_st to GT15_lag. Log transformed fold changes (log_2_FC) are plotted on the x-axis and significance (−log10FDR) is plotted on the y-axis. Red and blue denote genes with a significantly increased or decreased levels of expression (−1  ≥  log_2_FC  ≥  1, FDR  <  0.05), respectively, black indicates genes without significant change.

**Figure 2 microorganisms-10-01683-f002:**
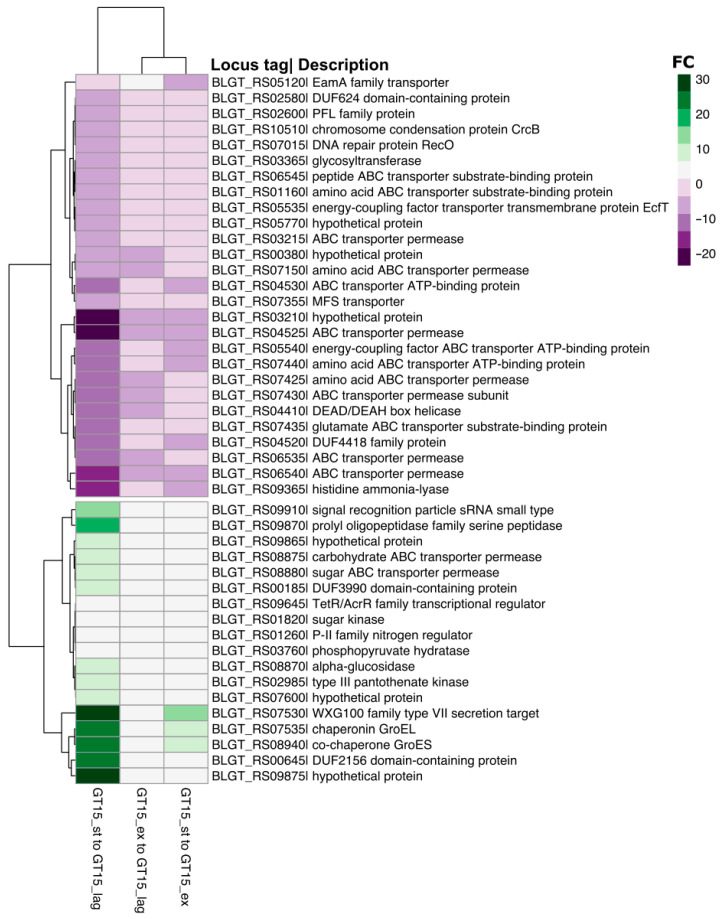
Comparison of 45 genes expressed in all the growth phases.

**Figure 3 microorganisms-10-01683-f003:**
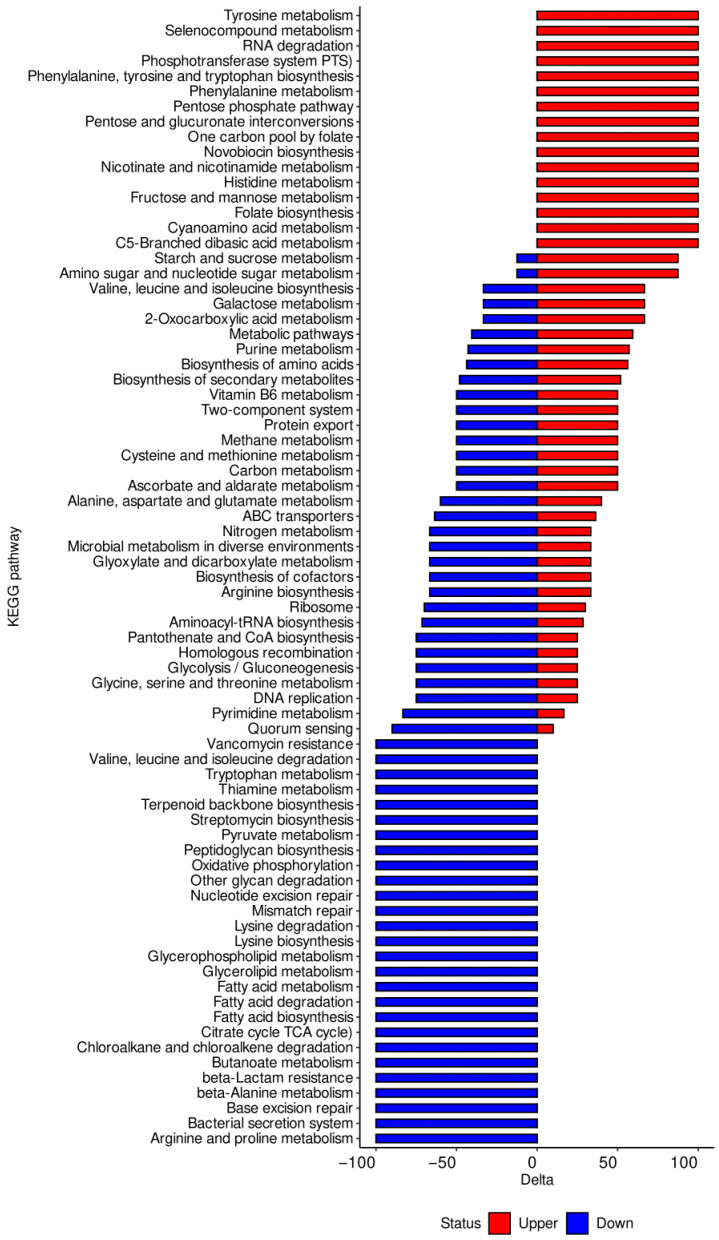
KEGG pathway enrichment analysis of DEGs (GT15_st compared to GT15_ex). The Y-axis represents the names of KEGG pathways. The X-axis represents the values of the Delta. Downregulated genes are shown in red. Upregulated genes are shown in blue. Delta is the relative distribution of upregulated and downregulated genes normalized with respect to the number of all differentially expressed genes of a certain KEGG pathway.

**Figure 4 microorganisms-10-01683-f004:**
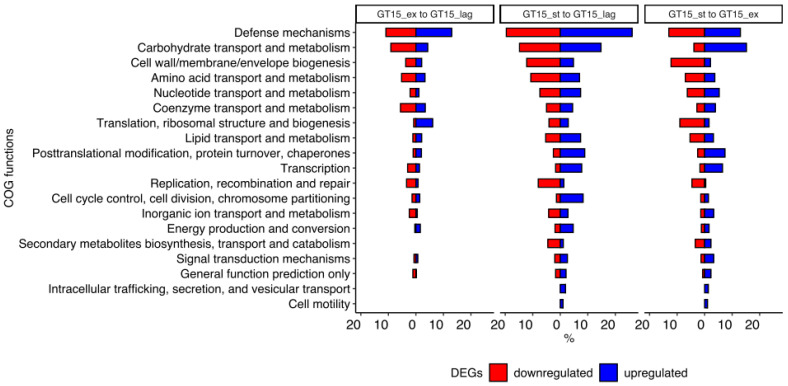
Functional analysis of DEGs throughout all the growth phases.

**Table 1 microorganisms-10-01683-t001:** Overall parameters of the generated RNA-Seq data.

Category	Growth Phase	Number of Raw Reads	GC, (%)	Average Length, bp	Number of Trimmed Reads	Number of Uniquely Mapped Reads	Percentage of Uniquely Mapped Reads (%)
Bl_K01_1	lag phase	4,114,356	60	101	4,104,837	3,684,561	89.80
Bl_K01_2	lag phase	4,084,494	60	101	4,072,394	3,639,926	89.40
Bl_K01_3	lag phase	3,795,673	60	101	3,786,859	3,380,989	89.30
Bl_K05_1	exponential phase	4,182,552	60	101	4,171,537	3,754,019	90.00
Bl_K05_2	exponential phase	3,983,507	60	101	3,972,238	3,566,746	89.80
Bl_K05_3	exponential phase	3,879,511	60	101	3,870,191	3,473,633	89.80
Bl_K09_1	stationary phase	4,470,198	59	101	4,463,213	4,007,847	89.80
Bl_K09_2	stationary phase	4,161,821	59	101	4,148,602	3,668,915	88.40
Bl_K09_3	stationary phase	3,881,059	59	101	3,867,621	3,419,213	88.40

## Data Availability

The dataset of RNA-Seq analysis was deposited to the NCBI under the project name PRJNA628664.

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
