# Peer review of "The Gene Expression Profile Differs in Growth Phases of the *Bifidobacterium Longum* Culture"

_microorganisms, 2022, doi:10.3390/microorganisms10081683_

Round 1

Reviewer 1 Report

The authors in the manuscript titled "Transcriptome analysis of growth phases in Bifidobacterium longum GT-15" have used RNA sequencing to profile differential gene expression in different growth stages for the gut microbe B. longum. The methods are well described, and the research design is well thought of and structured well. Assessing the differential expression profile for a gut microbe during different growth phase is definitely a relevant research question. In this manuscript the authors have written a comprehensive result and discussion section, but the content and display items (figures and Tables) lack clarity which needs to be addressed. Authors need to bring in more details in these sections in order to bring focus to what the DEGs means at the metabolic and stress response level for the microorganism itself. This can be addressed in a reworded revised version of the manuscript. Following are my comments:

1) In order to better understand and visualize the differential levels of gene expression the data presented in Table 2 and Table 3 need to be represented in a heatmap and bar graph respectively. This would help visualize the data better and understand the interpretations presented in the result and dicussiso section. Figure 2 needs to be separated in order to make the y-axis legible. You might show one of the three as a main figure and provide the other two as supplementary figures.

2) Authors mention metabolic pathways were upregulated as well as downregulated. KEGG metabolic pathways is a superpathway which includes all possible metabolic pathways and is too generic to include in the results section as is. It would be better to rewrite what specific metabolic pathways were downregulated versus upregulated. Similarly for two-component system. ABC transporters come up as upregulated and downregulated. Are there different types of ABC transporters that are uniquely upregulated versus downregulated?

3)    Authors also mention mitochondrial biogenesis, bacterial systems don’t have mitochondria. Authors need to manually curate the data they procured from the KEGG pathway and functional annotation of DEGs analysis to eliminate such discrepancies.

4) Authors mention in the stationary phase the expression of the BLGT_RS09365 gene that encodes a histidine ammonia-lyase dropped significantly. More information is required in the main text here, by how many fold, compared to which growth phase? Also refer to Table 2 at the end of this sentence. Similar writing style should be used throughtput the results section to provided the fold change, compared to which growth phase and finally refer to the Table that represents the data. This bring clarity to the results that the authors are reporting.

5)    Supplementary data is missing – Table S1, Table S2

6)   Was the RNA Seq data submitted to a major repository? (e.g. GEO, SRA, etc.)

7)   Authors mention that genes that were significantly differentially expressed (based on a fold change of at least two [log 2 ratio of >|2|] and a t-test P-value < 0.001) in each of the growth phases were counted. The statistical details should be mentioned separately in the methods section. Volcano plots (Fig 1) should show these significantly different DEGs as colored (blue and red) based on the criteria used in this manuscript instead of DEGs with a log 2 ratio of >= |1|.

8)   Conclusion and abstract should highlight key results - specific metabolic pathways/subsystems that were found to be differentially expressed in the different growth phases. This is not clearly mentioned in either section. Please rewrite these sections to reflect these key results.

Author Response

1) In order to better understand and visualize the differential levels of gene expression the data presented in Table 2 and Table 3 need to be represented in a heatmap and bar graph respectively. This would help visualize the data better and understand the interpretations presented in the result and dicussiso section. Figure 2 needs to be separated in order to make the y-axis legible. You might show one of the three as a main figure and provide the other two as supplementary figures.
Thank you for your comment. We added the figures.
2) Authors mention metabolic pathways were upregulated as well as downregulated. KEGG metabolic pathways is a superpathway which includes all possible metabolic pathways and is too generic to include in the results section as is. It would be better to rewrite what specific metabolic pathways were downregulated versus upregulated. Similarly for two-component system. ABC transporters come up as upregulated and downregulated. Are there different types of ABC transporters that are uniquely upregulated versus downregulated?
Thank you for your comment. In the results, we list the pathways. In the discussion, we discuss the main KEGG pathways in more detail. We added a new figure and a table with complete information on KEGG to the supplementary.
3) Authors also mention mitochondrial biogenesis, bacterial systems don’t have mitochondria. Authors need to manually curate the data they procured from the KEGG pathway and functional annotation of DEGs analysis to eliminate such discrepancies.
We apologize for this discrepancy. We checked all KEGG pathway and information in the text. We added a table with the main KEGG pathway to the supplementary.
4) Authors mention in the stationary phase the expression of the BLGT_RS09365 gene that encodes a histidine ammonia-lyase dropped significantly. More information is required in the main text here, by how many fold, compared to which growth phase? Also refer to Table 2 at the end of this sentence. Similar writing style should be used throughtput the results section to provided the fold change, compared to which growth phase and finally refer to the Table that represents the data. This bring clarity to the results that the authors are reporting.
Thank you for your comment. We have added information about changing expression
5) Supplementary data is missing – Table S1, Table S2
Thank you for your comment. We added the supplementary.
6) Was the RNA Seq data submitted to a major repository? (e.g. GEO, SRA, etc.)
Yes. The dataset of RNA-Seq analysis was deposited to the NCBI under the project name PRJNA628664. This information is presented in materials and methods (RNA Extraction, library preparation and high-throughput sequencing).
7) Authors mention that genes that were significantly differentially expressed (based on a fold change of at least two [log 2 ratio of >|2|] and a t-test P-value < 0.001) in each of the growth phases were counted. The statistical details should be mentioned separately in the methods section. Volcano plots (Fig 1) should show these significantly different DEGs as colored (blue and red) based on the criteria used in this manuscript instead of DEGs with a log 2 ratio of >= |1|.
We apologize for this discrepancy. We used the following criteria for DEGs: log2FC ≥ |1| (FC ≥ |2|) and FDR < 0.05. The typo in the "Results" section has been fixed: “based on a fold change of at least two [log 2 ratio <−2 or >2]” to “log2FC ≥ |1|, FDR < 0.05”. Volcano plots (Fig 1) and the remaining analysis are also based on these criteria.  
The  typo in the "Results" section has been corrected and the "Data processing and Analysis" subsection has been expanded.

8)   Conclusion and abstract should highlight key results - specific metabolic pathways/subsystems that were found to be differentially expressed in the different growth phases. This is not clearly mentioned in either section. Please rewrite these sections to reflect these key results.
Thank you for your comment. We added information to the Abstract and Conclusion.

Reviewer 2 Report

Dear authors,

Manuscript microorganisms-1811660 entiteled "Transcriptome analysis of growth phases in Bifidobacterium longum GT-15 and authored by Vladimir Veselovsky , Marina Dyachkova , Dmitry Bespiatykh , Roman Yunes , Egor Shitikov , Polina Polyaeva , Valeriy Danilenko , Evgenii Olekhnovich and Ksenia Klimina targets a hot topic that could be of high interest to the journal readers and to the scientific community as a whole. The idea is suitable and the paper suits well with the scope of the journal. The experiments seems to be nicely designed and conducted. Unfortunately few issues needs authors attention before the journal reaches the journal standards and can be formally accepted for publication:

1. The title "Transcriptome analysis of growth phases in Bifidobacterium longum GT-15" is not suitable. Please change it to a less neutral title that presents main findingy of the manuscript and that highlight the link between growth phases and life style of Bifidobacterium longum GT-15.

2. In the introduction section I have the feeling that the main focus of the paper have not been developped. Please discuss more the different environmental factors that Bifidobacterium longum GT-15 encounters in the human organism. Please highlight to the readers how these environmental factors are linked to the growth phases of Bifidobacterium longum GT-15. Your work have this aims so plaease discuss more this part !

3. In whole discussion only 9 references have been cited. I believe that main contributions to the field are not all cited please enrich your introduction I have the feeling that main aspects that needs to be highlighted are underlooked ! It is very important for the readers to have a well written introduction that presents state of the art research in the field!

4. Please in introduction discuss the main transcriptomic changes that are generally described for bacteria and mainly closely related bacteria to  Bifidobacterium longum GT-15.

5. Please document how the medium used was selected. Does this medium reflect some conditions that reming life style of Bifidobacterium longum GT-15? was it more relevant to test different media or this medium is enough to answer the questions that authors aim to address?

6. In results section : figure 2 is not readable ! please improve the resolution of this figure.

7. In the discussion part roughly 8 references have been used. This is not relevant Please enrich your discussion I feel that the main questions intended in this manuscript have not been discussed!

8. Please discuss your findings according to what is known about gene expression during the different growth phases.

9. There is no link to life style of Bifidobacterium longum GT-15 and gut conditions ! Please discuss and document more widely this part !

10. The conclusion part is very poor ! please highlight your findings and what this paper particularily brought to the field ! It seems to me that your conclusions are not supported by your results please stick closely to your result there are many nice points found that could really be the conclusiopn to this report !

I encourage authors to address all these issues to meet the journal standards and i will be happy to read an improved version of this manuscript that I can recommand for publication.

Best regards

Author Response

  1. The title "Transcriptome analysis of growth phases in Bifidobacterium longum GT-15" is not suitable. Please change it to a less neutral title that presents main findingy of the manuscript and that highlight the link between growth phases and life style of Bifidobacterium longum GT-15.

We corrected the Title «The gene expression profile differs in growth phases of the Bifidobacterium longum culture»

2. In the introduction section I have the feeling that the main focus of the paper have not been developped. Please discuss more the different environmental factors that Bifidobacterium longum GT-15 encounters in the human organism. Please highlight to the readers how these environmental factors are linked to the growth phases of Bifidobacterium longum GT-15. Your work have this aims so plaease discuss more this part !

Thank you for your comment. We added information to the Introduction.

  1. In whole discussion only 9 references have been cited. I believe that main contributions to the field are not all cited please enrich your introduction I have the feeling that main aspects that needs to be highlighted are underlooked ! It is very important for the readers to have a well written introduction that presents state of the art research in the field!

Thank you for your comment. We added information with references to the Introduction.

4. Please in introduction discuss the main transcriptomic changes that are generally described for bacteria and mainly closely related bacteria to  Bifidobacterium longum GT-15.

Thank you for your comment. We added information to the Introduction.

5. Please document how the medium used was selected. Does this medium reflect some conditions that reming life style of Bifidobacterium longum GT-15? was it more relevant to test different media or this medium is enough to answer the questions that authors aim to address?

The medium described is traditionally used by us for the effective cultivation of Bifidobacteria. The addition of cysteine is also a standard technique that improves the cultivation of anaerobic bifidobacteria (e.g., Guo-wei, S., Zhe, J., Tao, Q., He, C., & Qi, M. (2012, February). Effect of ascorbic acid and cysteine hydrochloride on growth of Bifidobacterium Bifidum. In Proceedings of the 2012 International Conference on Convergence Computer Technology (pp. 339-342).). Since we did not pursue the goal of determining the effect of various media or their components on the growth of a bifidobacteria culture, but to simulate some changing environmental conditions, in particular, greater or lesser availability of nutrients in different phases of culture growth, we believe that the medium is enough to answer the questions that we aim to address.

6. In results section : figure 2 is not readable ! please improve the resolution of this figure.

Thank you. We edited the figures.

7. In the discussion part roughly 8 references have been used. This is not relevant Please enrich your discussion I feel that the main questions intended in this manuscript have not been discussed!

hank you for your comment. We added information with references to the discussion

8. Please discuss your findings according to what is known about gene expression during the different growth phases.

Thank you for your comment. We rewritten the conclusion. We added the main conclusions of our study.

9. There is no link to life style of Bifidobacterium longum GT-15 and gut conditions ! Please discuss and document more widely this part !

Thank you for your comment. We added information to the Introduction. The strain Bifidobacterium longum subsp. longum GT15 was isolated from the feces of a healthy adult inhabiting Central Russia and demonstrated good probiotic properties [Zakharevich, N. V.; Averina, O. V.; Klimina, K.M.; Kudryavtseva, A. V.; Kasianov, A.S.; Makeev, V.J.; Danilenko, V.N. Complete Genome Sequence of Bifidobacterium Longum GT15: Unique Genes for Russian Strains. Genome Announc. 2014, 2, 1348–1362, doi:10.1128/genomeA.01348-14.]

10. The conclusion part is very poor ! please highlight your findings and what this paper particularily brought to the field ! It seems to me that your conclusions are not supported by your results please stick closely to your result there are many nice points found that could really be the conclusiopn to this report !

Thank you for your comment. We added information to the Сonclusions.

Round 2

Reviewer 2 Report

Dear authors,

Thanks for addressing all my comments. I can now recommend your manuscript for publication.

Best regards

Author Response

Thank you for your comment